# Evaluation of Polycyclic Aromatic Hydrocarbons in Smoked Cheeses Made in Poland by HPLC Method

**DOI:** 10.3390/molecules27206909

**Published:** 2022-10-14

**Authors:** Magdalena Polak-Śliwińska, Beata Paszczyk, Mariusz Śliwiński

**Affiliations:** 1Department of Commodity Science and Food Analysis, Faculty of Food Science, University of Warmia and Mazury in Olsztyn, Plac Cieszyński 1, 10-726 Olsztyn, Poland; 2Dairy Industry Innovation Institute Ltd., Kormoranów 1, 11-700 Mrągowo, Poland

**Keywords:** European Union priority polycyclic aromatic hydrocarbons, smoked cheeses, organic contaminants, food analysis

## Abstract

Smoked cheeses are particularly popular among consumers for their flavor and aroma. Of interest, therefore, is the health aspect related to the likelihood of polycyclic aromatic hydrocarbons (PAHs), which are known carcinogens found in smoked products. Thus, the purpose of this study was to evaluate the occurrence of 15 polycyclic aromatic hydrocarbons (PAHs) in smoked and non-smoked cheeses purchased in Poland to monitor their safety. The level of selected PAHs in cheese samples was determined using the HPLC-DAD-FLD method. Most of the cheeses tested met the maximum level of benzo[a]pyrene (2 μg/kg) and the sum of benz[a]anthracene, chrysene, benzo[b]fluoranthene and benzo[a]pyrene (12 μg/kg) established for these products. However, all the cheeses studied in this work had relatively low amounts of the sum of these compounds compared to the information available in the cheese literature, ranging from <LOD to 24.5 μg/kg. This amount does not pose a health risk to consumers. The predominant PAHs found were naphthalene, phenanthrene, fluorene and acenaphthene. Benzo[a]pyrene, the marker compound representing carcinogenic PAHs, was found in 100% and 0% of Polish smoked and non-smoked cheeses, respectively. Although there are currently no regulations for smoked cheeses and maximum concentrations of PAHs in this type of food product, control of PAHs content in cheeses is important due to the mutagenic and carcinogenic potential of these chemicals.

## 1. Introduction

Polycyclic aromatic hydrocarbons (PAHs) are a group of organic compounds with a structure consisting of carbon and hydrogen atoms that have more than two fused aromatic rings. Most PAHs have carcinogenic effects on animal or humans and induce various cancers. They are given priority concern because of their mutagenic and carcinogenic effects [1,2,3]. Polycyclic aromatic hydrocarbons are a consequence of environmental pollutants, imperfect burning or pyrolysis of organic substances during industrial processing [4,5,6]. Additionally, foods can be contaminated during their processing and preparation through different heat treatments. The carcinogenicity of PAHs varies from the potent to moderately carcinogenic PAHs which include 3-methylcholanthrene, benzo[a]pyrene, dibenz[a,h]anthracene, 5-methylchrysene, and dibenz[a,j]anthracene, whereas benzo[e]pyrene, dibenz[a,c]anthracene, chrysene, benzo[c]phenanthrene and fluoranthene are relatively weak or inactive carcinogens [7]. PAHs are classified according to the number of carbon rings into “heavy PAHs” with five or more aromatic rings or “light PAHs” with less than five rings [8]. Cooking processes have been found to be a major source of PAHs in foods. Although PAHs can also be formed during curing and processing of raw food prior to cooking, several researchers in recent years have shown that the major dietary sources of PAHs are fish and meat especially where there is high consumption of meat cooked over an open flame [7]. Several procedures and methods have been developed recently to assess and detect PAHs in foods and more recently, bio-monitoring procedures have also been developed to assess human exposure to PAHs [7]. Numerous organizations such as the United States Environmental Protection Agency (US EPA), the International Agency for Research on Cancer (IACR), the Scientific Committee on Food (SCF), the Joint FAO/WHO Expert Committee on Food Additives (JECFA), the International Programme on Chemical Safety (IPCS), and the European Food Safety Authority (EFSA) have been involved in evaluating the occurrence and toxicity of PAHs [7].

The SCF reviewed the presence and toxicity of PAHs in food and issued an opinion on 4 December 2002 [9]. The SCF concluded that benzo[a]pyrene may be used as a marker of occurrence and effect of the carcinogenic PAHs in food [10]. Afterward, the Commission asked Member States to monitor PAHs (and, in particular, the 15 priority substances identified by SCF as potentially genotoxic and carcinogenic to humans) in the Commission Recommendation 2005/108/EC [11,12]. In 2008, to highlight the strongly carcinogenic PAHs, the European Union (EU) has defined 16 priority pollutants (15 and 1 EU priority PAHs) that should commonly be monitored in foods [13,14]. In general, heavy PAHs tend to be more stable and toxic than lighter ones [15].

The EFSA collected the data submitted in the framework of this recommendation and issued a report where it was stated that the conclusion made by the SCF that benzo[a]pyrene is a good indicator for PAHs occurrence could not be demonstrated by the monitoring data from the Member States [10]. This statement was based on the fact that benzo[a]pyrene was not detected in about 30% of the samples where other PAHs among the 15 SCF priority ones were detected. Therefore, benzo[a]pyrene alone is not considered a suitable indicator for the occurrence and toxicity of genotoxic and carcinogenic PAHs [10,16]. PAHs have been monitored in a variety of foodstuffs, i.e., in traditionally smoked cheeses [17], in milk and other dairy products [18], in fruits and vegetables [19], in fried meats and fish [20,21,22], in fried foods based on wheat flour [23,24] and in vegetable oils [25,26,27,28]. Taking into consideration the sources of PAHs generation, adequate process and quality control of the processed foods could be a veritable mean to reduce PAHs ingestion in foods. The results of Sun at al. [29] indicated that the number of studies on PAHs in foods has been growing at an average annual rate of 13% [29]. The high yield of PAHs-related research in Europe may be attributed to concerns related to olive oil and smoked meats. Besides, other studies included the formation of PAHs in food processing, the concentration of PAHs in edible oils and smoked products, and the GC/MS method for detecting PAHs. Based on articles we have analysed, it is worth mentioning that determination of PAHs and derivatives still needs improvement. Moreover, with collective efforts from all researchers, there is hope that more progress will be made in the limit standards and toxicological evaluation for PAHs derivatives, as well as methods for the green removal of PAHs in food [29]. 

The purpose of this research was to determine the occurrence of European priority polycyclic aromatic hydrocarbons (EU PAHs) in smoked and unsmoked cheeses purchased on the Polish market in order to control their safety.

## 2. Results and Discussion

As mentioned above, Table 1 shows that the most abundant compounds in the studied Polish cheeses are the light PAHs, among which naphthalene, acenaphthylene, or phenanthrene can be cited. This correlates well with the results of previous studies on the occurrence of PAHs in other types of European smoked cheeses [30,31,32]. Recently, the European Food Safety Authority Panel on Contaminants in the Food Chain suggested that four PAHs (sum of benzo[a]pyrene, chrysene, benzo[a]anthracene, and benzo[b]fluoranthene) is the most suitable indicator for overall evaluation of PAHs level in food [16]. The maximum values for different groups of food (oils and fats, cocoa beans, coconut oil, meats and smoked fish, smoked sprats, molluscs, cereals, infant food, and infant formula) have been established through the European Communities Regulation 835/2011 [16]. In 2015, amending Regulation (EC) No 1881/2006 as regards maximum levels for polycyclic aromatic hydrocarbons in cocoa fibre, banana chips, food supplements, dried herbs, and dried spices have been established through the European Communities Regulation 2015/1933 [33,34]. The EFSA panel evaluated multiple substance groupings (PAH_2_ [B(a)P and Chr], PAH_4_ [B(a)P, Chr, B(a)A, and B(b)F] and PAH_8_ [B(a)P, Chr, B(a)A, B(b)F, B(k)A, B(ghi)P, DB(ah)A, and I(1,2,3-cd)P] for an exposure calculation to determine safe margins for exposure. The measured concentrations of PAHs in smoked cheeses are in line with previous findings reported by other research groups [35,36,37,38]. 

For smoked cheese samples, the concentrations of B[*a*]A, Chr, B[*b*]F, and B[*a*]P as PAH_4_ in the present study were ND–2.2 µg/kg, ND–0.9 µg/kg, ND, and <LOD–0.9 µg/kg, respectively. The concentrations of PAHs measured in a study performed by Suchanová et al. [39] were similar to those found in this study.

A big part of analysed compounds in smoked and unsmoked cheese samples were Phe, Fln, Naph and Ace while the others were present at relatively low levels. The sum of 9 PAHs (Ace, Ant, B[*a*]A, B[*k*]F, B[*ghi*]P, B[*a*]P, DB[*ah*]A, Naph, and Pyr) and sum of 5 carcinogenic PAHs (B[*a*]A, B[*k*]F, B[*ghi*]P, B[*a*]P, DB[*ah*]A) are presented in Table 1. Results showed that sum of 9 PAHs and 5 carcinogenic PAH concentrations in smoked cheese samples ranged from 8.9 to 25.8 μg/kg and from 0.7 to 2.9 μg/kg, respectively. The sum of 9 PAHs concentrations in unsmoked cheese samples ranged from 9.9 to 19.4 μg/kg. The sum of 5 carcinogenic PAHs in unsmoked cheese samples was not detected. Suchanová et al. [40] demonstrated that the sum of 12 PAHs was significantly lower in commercial smoked cheese (2.3 to 57 μg/kg) than in home-made cheese (73 to 114 μg/kg). Higher contamination may be attributed to the deposition of solid particles (from smoke during the smoking process) containing PAHs on the cheese surface [39]. Additionally, in the traditional smoking treatment, the smoke comes in direct contact with cheese surface due to uncontrolled conditions [40]. In the industrial process, smoke used is purified from hazardous compounds of wood pyrolysis, and controlled production conditions are provided for the duration of cheese smoking [39]. 

B[*a*]P is the most important among PAHs in all of the food samples because of its toxicity [2,3,41,42]. The B[*a*]P contents of traditional and industrial smoked cheeses were determined as 0.69 and 0.25 μg/kg, respectively (*p* < 0.05). The B[*a*]P contents of Diavoletto cheese (a typical Italian smoked cheese) smoked with a mix of wood shavings were 0.1 and 0.28 μg/kg in semi-industrial and traditional cheeses, respectively [40]. The B[*a*]P contents determined in both traditional and industrial unsmoked Circassian cheese samples were clearly lower than those of smoked cheese.

The B[*a*]P levels in all cheese samples were lower than the maximum tolerable limits (5 μg/kg for smoked meat) set by Commission Regulation (EC) No. 1881/2006 dated 19 December 2006 [33]. The concentrations of B[*a*]P reported for similar smoked cheese from Italian, Spanish, and Czech markets were also lower than this limit [32,38,39,40,41,43,44,45].

The method validation parameters are presented in Table 2. The LODs of individual analytes ranged from 0.05–0.10 µg/kg (Table 2). The lowest LOD was found for B[*ghi*]P (0.04 µg/kg), while the other compounds had higher LODs at 0.05–0.10 µg/kg. The linearity test showed that for each compound tested, the R^2^ value was greater than or equal to the critical value of 0.9995. The coefficient of variation (CV %) value for all analytes was ≤0.9% (Table 3). The highest CV was observed for B[*a*]F (0.9%), while for the other compounds it was in the range of 0.6–0.8%. The LOQ was 0.15 µg/kg. Recovery averaged 94.10% (level of 5.0 µg/kg) and 87.20 (level of 50.0 µg/kg). The example HPLC chromatogram is presented in Figure 1.

## 3. Materials and Methods

### 3.1. Cheese Samples

For this study, the materials were commercial smoked (*n* = 10) and unsmoked (*n* = 5) cheese samples produced with cow milk. Smoked and unsmoked cheese samples were randomly acquired in a local market or industrial source (Table 3). The samples came from various Polish producers: SM Mlekpol (samples Nos. SC 1–4, Grajewo, Poland), Hochland Polska Ltd., (sample No. SC-5, Węgrów, Poland), SM Mlekovita (samples Nos. SC 6–7, Wysokie Mazowieckie, Poland), SM Spomlek (sample No. SC-8, Radzyń Podlaski, Poland), OSM Włoszczowa (sample No. SC-9, Włoszczowa, Poland) and OSM Sierpc (sample No. SC-10, Sierpc, Poland). The unsmoked cheese products (sample Nos. UC 1–5) obtained from the common market were used as the reference samples. When supplied, samples were stored at −18 °C for short duration until analysis.

### 3.2. Chemicals and Reagents

The standards of PAHs were purchased from Sigma (Sigma Aldrich, Steinheim, Germany). Water was purified with a Milli-Q ultra-pure water system (Millipore, Bedford, MA, USA) with high resistivity (>18.0 M·cm) throughout the experiments. Acetonitrile, cyclohexane, dichloromethane, potassium hydroxide, ethanol and methanol (all of HPLC grade, 99–99.9%) were supplied by Merck (Darmstadt, Germany). Solid-phase extraction (SPE) was performed with Discovery^®^ DSC-18 (silica 500 mg/3 mL) cartridges obtained from Supelco (Bellefonte, PA, USA). Mobile phase was filtered through a Millipore 0.22 µm membrane filter before use.

### 3.3. PAHs Standards

The PAHs standards used were the following: a commercial mixture of PAHs standards–PAHs Calibration Mix (*Trace*CERT^®^, certified reference material, 10 µg/mL each component in acetonitrile from Supelco, Bellefonte, PA, USA), containing 16 priority PAHs: acenaphthene (Ace), fluorene (Fln), anthracene (Ant), pyrene (Pyr), benz[*a*]anthracene (B[*a*]A), chrysene (Chr), benzo[*b*]fluoranthene (B[*b*]F), benzo[*k*]fluoranthene (B[*k*]F), benzo[a]pyrene (B[*a*]P), indeno [1,2,3-*cd*]pyrene (I [1,2,3-*cd*]P), dibenz[*ah*]anthracene (DB[*ah*]A), benzo[*ghi*]perylene (B[*ghi*]P), phenanthrene (Phe), fluoranthene (Flt), naphthalene (Naph). This Certified Reference Material (CRM) was produced and certified in accordance with ISO 17034 [46] and ISO/IEC 17025 [47]. 

Working standard solutions (concentrations in the range 0.01, 0.1, 1.0, 2.0, 5.0, 10.0, 20.0, 50.0 and 70.0 ng/mL) were prepared in acetonitrile and stored at −20 °C. Before use, all glassware was washed with detergent, rinsed with distilled water and acetone and then dried at 220 °C. All the above-mentioned standards were used in the identification and quantification of the PAHs present in the samples.

### 3.4. Determination of PAHs Content

#### 3.4.1. Extraction and Clean-Up

PAHs were measured by HPLC-FLD method according to Anastasio et al. [35] with some modifications. Cheese samples were prepared after a cleaning procedure. For this purpose, essentially, 1 g cheese sample after homogenization was weighed into 50 mL teflon centrifuge tubes, and 5 mL of 1 M KOH ethanolic solution was added and then placed for 2 h in a water bath at 80 °C. After cooling to room temperature, 5 mL of distilled water and 10 mL of cyclohexane were added, and the mixture was vortexed for 10 min. After centrifugation at 4000 *g* for 15 min, the supernatant layer was re-extracted with 10 mL of cyclohexane as previously described. The two cyclohexane phases were collected in a volumetric flask and concentrated by a rotary vacuum evaporator at 40 °C, followed by drying of the extract under a nitrogen stream. The residue was dissolved in 2 mL of acetonitrile, which was applied to an SPE Discovery^®^ DSC-18 cartridge. The clean-up method used in this study was based on the method described by Węgrzyn et al. [36]. The SPE cartridge was previously activated by the passage of 10 mL of ultrapure water and 10 mL of methanol. The cartridge was then dried. After loading the 2 mL of eluate, the cartridge was left to dry by air for 1 min, and PAHs were eluted with 10 mL of dichloromethane. Prior to HPLC analysis, the collected eluate was evaporated to near dryness in a water bath (40 °C) under a nitrogen stream. Finally, the PAHs fraction was diluted with 1.5 mL acetonitrile before the HPLC-DAD-FLD determinative step, and the solution was filtered using a 0.45 µm membrane syringe filter. This solution was transferred into a 2 mL amber vial.

#### 3.4.2. Chromatographic Analysis

The extracts of PAHs were analysed using an HPLC system from Shimadzu LC-10A (Kyoto, Japan), consisting of a fluorescence detector (FLD) RF-10XL, a diode array detector (DAD) SPD-M20A, quaternary pump LC-20AT, degassing device DGU-20A, column oven CTO-10A and auto-injector SIL-10A. The HPLC column was a Supelcosil^®^ LC-PAH (250 mm × 4.6 mm i.d.; 5 µm) column with the guard column Supelcosil^®^ LC-18 (20 mm × 4.0 mm i.d., 5 µm; Supelco, Bellefonte, USA) to protect the analytical column that was used for the analysis of sample extracts. The HPLC conditions were the following: gradient elution (mobile phase: 0 min–55% acetonitrile +45% water, 20 min–100% acetonitrile, 30.1 min–55% acetonitrile) with a 1 mL/min flow rate. The monitoring of PAHs was performed by using a FLD detector (excitation/emission wavelength in nm) with the following FLD settings for the detection of PAHs and a diode array detector (DAD). The excitation and emission wavelengths were selected based on available literature. The separation parameters are shown in Table 4.

PAHs standard were injected into HPLC prior to any cheese sample (50 μL) injections. The PAHs profile in the sample was identified and quantified by the software program LC Solution (Shimadzu, Kyoto, Japan) by using the amount and peak area of standards and peak area of PAHs in the cheese sample, and the amount of PAHs in a cheese sample was estimated. The average amount of PAHs in all cheese samples was obtained and compared with the Polish standards. The external standard method was used for quantitative analysis. The quantitative and qualitative interpretation of the obtained chromatograms was carried out on the basis of the comparison of the retention time and the size of the area of PAHs peaks in standard samples of known concentration, retention time, and the size of the analyte peak area in the test samples.

#### 3.4.3. Recovery Studies

The HPLC-DAD-FLD method for the quantification of PAHs was validated for parameters such as linearity, limit of quantification (LOQ), limit of detection (LOD) and recovery. Linearity was determined by regression analysis, where calibration curves for individual PAHs were constructed by plotting the average peak area against concentration and generating a regression equation. The limits of LOD and LOQ were defined as the lowest concentration of the sample determined by the analytical method to obtain signal-to-noise ratios of 3:1 and 10:1, respectively. LOD and LOQ were calculated using the following formulas: LOD = Cm + 3SD and LOQ = Cm + 6SD, where Cm is the mean value of the PAH concentration in the blank, and SD is the standard deviation. The measurement procedure included preparation and analysis of blank samples with unknown PAH concentrations and two samples containing calibration solutions (5.00 and 50.00 µg/kg). The spiked and unspiked samples were analyzed under the same conditions to prevent matrix effects on peak positions on the chromatogram, as well as to evaluate the percentage recovery of PAHs. Results were expressed in micrograms per kilogram of cheese (µg/kg). Recovery values were determined by performing the entire analytical procedure for the determination of PAHs in smoked and unsmoked cheese samples labeled with a mixture of PAH standards at concentrations of 5.00 µg/kg and 50.00 µg/kg in triplicate. Based on the results, recovery values were calculated.

#### 3.4.4. Data Analysis

All analyses were conducted in triplicate and the data expressed as mean ± standard deviation. Data were subjected to analysis of variance (ANOVA), followed by the Duncan test for comparison of means. All statistical analyses were performed using Statistica 12.5 PL software (StatSoft, Kraków, Poland). All calculations were performed at a *p* ≤ 0.05 significance level.

## 4. Conclusions

This research paper highlights one of the major contaminants found in food processing, namely PAHs, which have been under monitoring for years. The ever-increasing number of articles on the presence of PAHs in food has attracted worldwide attention due to their ubiquity and the resulting health risks that can result from ingestion, inhalation and dermal contact with PAHs. Using HPLC-DAD-FLD, the presence of PAHs was confirmed in samples of smoked and non-smoked cheeses that raise health concerns. The cheeses contained trace amounts of benzo[a]pyrene. The low levels of benzo[a]pyrene and the sum of benzo[a]pyrene, benzo[a]anthracene, benzo[b]fluoranthene and chrysene in the analyzed cheeses were most likely due to mild smoking in warm smoke. This type of research is necessary to know the magnitude of harmful compounds that may be present in the tested products, which are popularly consumed by humans. The lack of clear regulations for smoked cheeses and maximum concentrations of PAHs does not change the fact that the control of PAHs in cheeses is necessary due to the mutagenic and carcinogenic potential of these compounds, and to ensure that the PAH content in food is maintained at a level that does not threaten the health of consumers.

## Figures and Tables

**Figure 1 molecules-27-06909-f001:**
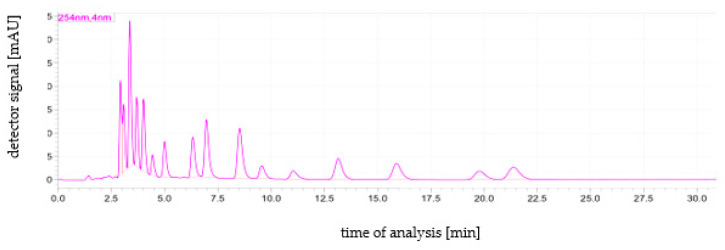
An example HPLC chromatogram showing the order of PAHs elution. Elution order: Nat, Ace, Fln, Phe, Ant, Flt, Pyr, B[*a*]A, Chr, B[*b*]F, B[*k*]F, B[*a*]P, DB[*ah*]A, B[ghi]P, I [1,2,3-*cd*]P. The sample was a calibration standard with a concentration of 50 ng/mL of each target PAH analyte.

**Table 1 molecules-27-06909-t001:** PAHs content in reference unsmoked and smoked cheese samples (*n* = 3; mean value) (µg/kg).

Sample Code	Sum of PAH_5_	Sum of PAH_9_	Naph	Ace	Fln	Phe	Ant	Flt	Pyr	B[*a*]A	Chr	B[*b*]F	B[*k*]F	B[*a*]P	DB[*ah*]A	B[*ghi*]P	I[1,2,3-*cd*]P
**UC-1**	ND	9.9	9.1	0.3	1.2	2.1	<LOD	<LOD	0.5	ND	ND	<LOD	<LOD	<LOD	ND	ND	ND
**UC-2**	ND	19.4	8.8	0.7	1.1	3.2	<LOD	<LOD	0.4	ND	ND	<LOD	<LOD	<LOD	ND	ND	ND
**UC-3**	ND	13.1	11.9	0.5	1.6	4.1	<LOD	<LOD	0.7	ND	ND	<LOD	<LOD	<LOD	ND	ND	ND
**UC-4**	ND	10.7	10.2	0.2	2.1	3.6	<LOD	<LOD	0.3	ND	ND	<LOD	<LOD	<LOD	ND	ND	ND
**UC-5**	ND	13.4	12.3	0.5	1.1	4.2	<LOD	<LOD	0.4	ND	ND	<LOD	<LOD	<LOD	ND	ND	ND
**SC-1**	0.7	8.9	2.1	1.2	5.6	15.4	1.1	2.2	1.1	0.1	0.2	ND	ND	0.6	ND	ND	ND
**SC-2**	0.9	10.6	3.7	1.1	5.0	11.7	4.3	3.1	1.3	0.2	0.1	ND	ND	0.7	ND	ND	ND
**SC-3**	0.7	21.9	11.3	5.6	5.3	17.3	3.7	1.3	1.2	0.1	0.3	ND	ND	0.6	ND	ND	ND
**SC-4**	0.7	11.8	5.1	3.7	10.1	8.6	1.2	1.7	1.7	0.1	0.1	ND	ND	0.6	ND	ND	ND
**SC-5**	1.3	15.1	9.4	2.3	4.8	8.8	1.7	1.3	1.4	0.2	0.1	ND	ND	0.9	ND	ND	ND
**SC-6**	1.8	25.8	8.5	8.6	1.1	4.1	4.6	3.7	2.8	1.3	0.5	ND	ND	0.5	ND	ND	ND
**SC-7**	1.4	19.0	2.1	8.7	1.1	18.2	5.8	3.1	1.3	1.1	0.9	ND	ND	0.3	ND	ND	ND
**SC-8**	2.2	14.6	1.5	2.7	1.2	7.3	5.6	4.2	3.1	1.7	0.5	ND	ND	0.5	ND	ND	ND
**SC-9**	2.8	21.2	3.8	2.1	11.8	4.1	8.4	3.4	2.6	2.1	0.5	ND	ND	0.7	ND	ND	ND
**SC-10**	2.9	16.8	2.9	7.1	4.9	24.5	1.2	5.7	3.4	2.2	0.7	ND	ND	0.7	ND	ND	ND
**Mean-UC** ±SD			**10.5 ^a^** ±0.53	**0.4 ^a^** ±0.11	**1.4 ^a^** ±0.13	**3.4 ^a^** ±0.21	-	-	**0.5 ^a^** ±0.11	-	-	-	-	-	-	-	-
min.			8.8	0.2	1.1	2.1	-	-	0.3	-	-	-	-	-	-	-	-
max.			12.3	0.7	2.1	4.3	-	-	0.7	-	-	-	-	-	-	-	-
**Mean-SC** ±SD			**5.0 ^b^** ±0.54	**4.3 ^b^** ±0.25	**5.1** ±0.56	**12.0 ^b^** ±0.64	**3.8** ±0.67	**3.0** ±0.51	**3.2 ^b^** ±0.63	**0.9** ±0.11	**0.4** ±0.10	-	-	**0.6** ±0.13	-	-	-
min.			1.5	1.1	1.1	4.1	1.1	1.3	1.1	0.1	0.1	-	-	0.3	-	-	-
max.			11.3	8.7	11.8	24.5	8.4	5.7	3.4	2.2	0.9	-	-	0.9	-	-	-

Explanation: ND–not detected, ±SD–standard deviation, <LOD–below the limit of detection (LOD values in Table 2), ^a,b^—mean values with the different letter are significantly different at *p* ≤ 0.05.

**Table 2 molecules-27-06909-t002:** The method validation parameters.

Analyte	Recovery (%)		CV (%)	LOD (μg/kg)	LOQ (μg/kg)	Linearity R^2^
	Level I (5.0 μg/kg)	Level II (50.0 μg/kg)				
**Naph**	82.50	90.40	0.7	0.06	0.15	0.9998
**Ace**	93.40	83.80	0.8	0.06	0.15	0.9997
**Fln**	85.20	85.10	0.7	0.06	0.15	0.9996
**Phe**	90.50	80.20	0.6	0.10	0.15	0.9995
**Ant**	93.30	88.30	0.7	0.08	0.15	0.9998
**Flt**	86.40	87.60	0.8	0.06	0.15	0.9997
**Pyr**	92.60	88.50	0.7	0.10	0.15	0.0006
**B[*a*]A**	87.50	90.10	0.7	0.06	0.15	0.9997
**Chr**	80.30	92.50	0.6	0.05	0.15	0.9996
**B[*b*]F**	81.60	82.10	0.9	0.08	0.15	0.9996
**B[*k*]F**	92.70	84.80	0.6	0.05	0.15	0.9995
**B[*a*]P**	90.10	91.50	0.8	0.05	0.15	0.9997
**DB[*ah*]A**	88.20	90.80	0.7	0.05	0.15	0.9998
**B[*ghi*]P**	90.60	85.10	0.6	0.04	0.15	0.9998
**I[1,2,3-*cd*]P**	84.20	87.60	0.7	0.07	0.15	0.9996

Explanation: LOQ–limit of quantification, LOD–limit of detection (based on a S/N = 3) were determined using PAH standard mixtures in acetonitrile injected directly onto the HPLC column, CV–coefficient of variation.

**Table 3 molecules-27-06909-t003:** The origin of smoked and unsmoked cheese samples examined in the present study.

Sample Code ^1^	Category of Cheese Producer/Source	Cheese Commercial Name	Weight Of Single Cheese Package (g)
UC-1	Samples collected from retailed market	Gouda	250
UC-2		Edam	250
UC-3		Gouda	150
UC-4		Gouda	150
UC-5		Ser królewski	230
SC-1	Industrial	Gouda wędzona	1500
SC-2	Industrial	Salami wędzone	1000
SC-3	Industrial	Rolada ustrzycka	500
SC-4	Industrial	Salami królewskie	500
SC-5	Samples collected from retailed market	Gouda wędzona	250
SC-6		Zakopiańskie specjały–mini gołka zakopiańska	160
SC-7		Rolada ustrzycka	300
SC-8		Radamer wędzony	250
SC-9		Włoszczowski wędzony	250
SC-10		Ser królewski wędzony	200

^1^ Sample code: UC-unsmoked cheeses, SC-smoked cheeses.

**Table 4 molecules-27-06909-t004:** Separation parameters of HPLC-DAD-FLD.

Chromatografic Condition	
Parameter	Value
Analytical column	Supelcosil^®^ LC-PAH (250 mm × 4.6 mm i.d.; 5 µm) column with the guard column Supelcosil^®^ LC-18 (20 mm × 4.0 mm i.d., 5 µm; Supelco)
Mobile phase/Gradient	(A) Water; (B) Acetonitrile;
	0 min—55% B at 1 mL/min 5 min–55% B20 min–100% B 30 min–100% B 30.1 min–55% B
Injection volume	50 µL, needle washed for 3 s with acetonitrile
Temperature of the column	35 °C
Diode Array Detector (DAD)	254 nm, band width 4 nm, reference 400 nm, reference band width 100 nm, 10 Hz
Fluorescence Detector (FLD)	Multisignal acquisition, T1 = 216/336 for 7.1–10.7 min (Naph); T2 = 240/320 for 10.7–11.1 min (Ace, Fln); T3 = 248/368 for 11.1–12.2 min (Phe); T4 = 248/404 for 12.2–13.2 min (Ant); T5 = 232/448 for 13.2– 14.3 min (Flt); T6 = 270/388 for 14.3–16.0 min (Pyr); T7 = 270/388 for 16.0–19.3 min (B[*a*]A, Chr); T8 = 250/430 for 19.3–23.6 min (B[*b*]F, B[*k*]F, B[*a*]P); T9 = 295/405 for 23.6– 25.8 min (DB[*ah*]A, B[*ghi*]P); T10 = 248/484 for 25.8–30.5 min (I [1,2,3-*cd*]P;19.45 Hz; PMT 10

## Data Availability

Data is contained within the article.

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
