# Peer review of "Evaluation of Polycyclic Aromatic Hydrocarbons in Smoked Cheeses Made in Poland by HPLC Method"

_molecules, 2022, doi:10.3390/molecules27206909_

Round 1
Reviewer 1 Report
The present manuscript entitled "Evaluation of PAH in smoked cheeses made in Poland by HPLC with fluorescence detection" by Magdalena Polak-Śliwińska *, Beata Paszczyk, and Mariusz Władysław Śliwiński (molecules-1926540) is written correctly and has a good structure; moreover, it has all the necessary parts. The article is interesting from an analytical point of view; therefore, it should interest the reader. I proposed improvements in the method description and with a presentation of figures. The paper meets Molecules' requirements, and I recommend the article for publication in Molecules following the common editing stage. My current decision is a major revision. More specific comments and observations are presented below.
1. Title. Please specify which PAHs. EPA or EU? The same for Abstract.
2. The following statements can be found in the article "EUPAHs" and "EU PAHs". Please, standardize the record.
3. Please check if all abbreviations used are explained before use.
4. Page 5, line 165; page 7, line 229. PHA or PAHs?
5. Section 3.2. What were the parameters of the water used?
6. Section 3.2. Please add information in what form PAHs were purchased.
7. Only a mix was used, or the standards of single substances too?
8. Page 6, line 182. Please add information about specific concentrations, not a range.
9. Section 3.4.1. Were the samples filtered?
10. On what basis were the excitation and emission wavelengths selected? Was literature used or experiments performed?
11. An exemplary chromatogram for the standard and test samples should be added.
12. What method was used for quantitative analysis: external calibration or standard addition method?
13. The method should be validated and the basic validation parameters calculated and reported. The calculated validation parameters should be collected in a separate table. And the validation process should be described in detail.
14. Was there a co-elution of other substances? What can be done in the event of strong interference effects? How would you deal with them? What types of interference effects could occur?
15. Table 1 should be better discussed in the text. How was LOD calculated? The data in Table 1 should be presented more clearly. With a mean value, it is better to enter SD than ranges.
16. Page 4, line 131. Indexes by units should be corrected.
17. Does the developed method have disadvantages?
18. The measurement parameters of the FLD detector should be given in more detail.
19. Conclusion. Please, emphasize clearly the advantages of the research carried out.
20. Appropriate tools should be used to best characterize the method when developing a new approach (e.g., RGB Additive Color Model to Analytical Method Evaluation).
I hope that the comments presented will help improve the article.
Author Response
Comments and Suggestions for Authors
The present manuscript entitled "Evaluation of PAH in smoked cheeses made in Poland by HPLC with fluorescence detection" by Magdalena Polak-Śliwińska *, Beata Paszczyk, and Mariusz Władysław Śliwiński (molecules-1926540) is written correctly and has a good structure; moreover, it has all the necessary parts. The article is interesting from an analytical point of view; therefore, it should interest the reader. I proposed improvements in the method description and with a presentation of figures. The paper meets Molecules' requirements, and I recommend the article for publication in Molecules following the common editing stage.
My current decision is a major revision. More specific comments and observations are presented below.
- Please specify which PAHs. EPA or EU? The same for Abstract.
We agree with the reviewer's comment, the correction has made in the text of the manuscript. In the abstract the name analytes have been presented more precisely, the information about it has been added.
- The following statements can be found in the article "EUPAHs" and "EU PAHs". Please, standardize the record.
The record has been standardized in the revised manuscript according to the suggestion.
- Please check if all abbreviations used are explained before use.
The care has been taken to ensure that all abbreviations used are explained before use according to the suggestion.
- Page 5, line 165; page 7, line 229. PHA or PAHs?
This correction was made in the revised manuscript according to the suggestion.
- Section 3.2. What were the parameters of the water used?
For all HPLC/ UHPLC applications, type I ultrapure water with high resistivity (>18.0 MΩ.cm), free of particulate matter, bacteria, organic and ionic compounds, should be used to ensure good chromatographic performance, so such this water, dedicated to the HPLC method, was used during the analyses. Deionised water was obtained from Milli-Q water purification system (Millipore, Bedford, MA, USA).
- Section 3.2. Please add information in what form PAHs were purchased.
The PAHs standards used were the following: a commercial mixture of PAHs standards - PAHs Calibration Mix (TraceCERT®, certified reference material, 10 µg/ml each component in acetonitrile from Supelco, Bellefonte, PA, USA).
- Only a mix was used, or the standards of single substances too?
The mixture of PAHs standards was used in the method.
- Page 6, line 182. Please add information about specific concentrations, not a range.
A 10 µg/ml standard mixture containing all target analytes was prepared by consecutive acetonitrile dilutions. Calibration standards of different concentrations (0.01, 0.1, 1, 2, 5 10, 20, 50, 70 ng/mL) were prepared in acetonitrile (for HPLC analysis).
- Section 3.4.1. Were the samples filtered?
Prior to HPLC analysis, the collected eluate was evaporated to near dryness in a water bath (40°C) under a nitrogen stream. Finally, the PAHs fraction was diluted with 1.5 mL acetonitrile before the HPLC-FLD determinative step and the solution was filtered using a 0.45 µm membrane syringe filter. This solution was transferred into a 2 mL amber vial. Mobile phases were filtered through a Millipore 0.22 µm membrane filter before use.
- On what basis were the excitation and emission wavelengths selected? Was literature used or experiments performed?
The excitation and emission wavelengths were selected based on available literature. This information has been added in the text of manuscript according of the suggestion.
- An exemplary chromatogram for the standard and test samples should be added.
The example chromatograms for the standard sample is included in the reviewed manuscript.
- What method was used for quantitative analysis: external calibration or standard addition method?
The external standard method (ESTD) was used for quantitative analysis. This information is included in the reviewed manuscript.
- The method should be validated and the basic validation parameters calculated and reported. The calculated validation parameters should be collected in a separate table. And the validation process should be described in detail.
We agree with the reviewer's comment, the corrections have made in the text of the manuscript.
- Was there a co-elution of other substances? What can be done in the event of strong interference effects? How would you deal with them? What types of interference effects could occur?
HPLC-UV and HPLC-FLD suffer from uncertainty of identification due to possible interference from other compounds. Benzo[e]pyrene is used as a reference PAH to assess temporal variability and degradation patterns of other PAHs in environmental media due to its stability. Control of the instruments, data acquisition and analysis were performed with Lab Solution LC software. Different types of interference effects can occur in laboratory practice: proportional (multiplicative) - changing the slope of the calibration plot, constant (additive) - making a constant contribution to the signal measured for the analyte, and nonlinear - described by various mathematical functions. In addition, combinations of these effects can also occur during chemical analysis. Compensating for interfering substances for analytes can be done by using an appropriate methodological approach to determine accurate results, such as using the Good Methodical Practice approach. Of the two most commonly used methods, external calibration (EC) and standard addition method (SAM), only the latter allows compensation for interference effects that occur. However, it should be emphasized that these effects are proportional. Over the years, new approaches have been developed that provide new opportunities to eliminate interference effects, as well as to assess accuracy. An example of such approaches is the integrated calibration method (ICM) in its basic variety, as well as in its version with a complementary dilution method (ICM/CDM). By comparing the results obtained by interpolation and extrapolation, multiplicative effects can be eliminated. At the same time, the dilution version of ICM/CDM partially compensates for the nonlinear dependence of the calibration.
- Table 1 should be better discussed in the text. How was LOD calculated? The data in Table 1 should be presented more clearly. With a mean value, it is better to enter SD than ranges.
We agree with the reviewer's comment, the corrections have made in the text of the manuscript.
- Page 4, line 131. Indexes by units should be corrected.
I agree with the reviewer's comment, the indexes have been corrected by unit according to the suggestion.
- Does the developed method have disadvantages?
The developed method has disadvantages related to the lack of use of the standard addition method and/or internal standard. On the other hand, the disadvantages of the method are the specific isolation of the analyte from the complex food matrix, overlapping interference with the analytical image, which was associated with the need for a much larger number of variants of the experiment to resolve these issues fully.
- The measurement parameters of the FLD detector should be given in more detail.
This information has been added in text of manuscript according of the suggestion (in the form of added table).
- Conclusion. Please, emphasize clearly the advantages of the research carried out.
This research paper highlights one of the major contaminants found in food processing, namely PAHs, which have been under monitoring for years. The ever-increasing number of articles on the presence of PAHs in food has attracted worldwide attention due to their ubiquity and the resulting health risks that can result from ingestion, inhalation and dermal contact with PAHs. Using RP-HPLC/FLD, the presence of PAHs was confirmed in samples of smoked and non-smoked cheeses that raise health concerns. The cheeses contained trace amounts of benzo[a]pyrene. The low levels of benzo[a]pyrene and the sum of benzo[a]pyrene, benzo[a]anthracene, benzo[b]fluoranthene and chrysene in the analyzed cheeses were most likely due to mild smoking in warm smoke. This type of research is necessary to know the magnitude of harmful compounds that may be present in the tested products, which are popularly consumed by humans. The lack of clear regulations for smoked cheeses and maximum concentrations of PAHs, does not change the fact that the control of PAHs in cheeses is necessary due to the mutagenic and carcinogenic potential of these compounds, and towards ensuring that the PAH content in food is maintained at a level that does not threaten the health and life of consumers.
- Appropriate tools should be used to best characterize the method when developing a new approach (e.g., RGB Additive Color Model to Analytical Method Evaluation).
Thank you very much for your valuable suggestion. Recently developed and made available by a group of scientists in Krakow (Poland), a new way to more holistically evaluate analytical methods, the RGB (red-green-blue) additive color model provides a way to more fully evaluate the effectiveness, safety/sustainability and practicality of an analytical method. In addition to the usual measures of a method's analytical performance: accuracy, precision, linear range, detection limit, specificity, etc., there are guidelines for these evaluations, which are commonly provided by a number of governing bodies (e.g., the Environmental Protection Agency, EPA or the Food and Drug Administration, FDA), depending on the application associated with the method. However, other aspects of the method, such as low energy consumption, so important nowadays, low occupational risk, and minimal or no risk associated with reagents and waste, are also desirable features of any process (concepts in line with "green analytical chemistry"). Time and cost savings, reduced process and equipment complexity, low susceptibility to failure and robust performance are sought after. In the face of these issues, the RGB model provides a new way to evaluate these important aspects of the analytical method. Analytical performance is red, the "greenness" of the method is green, and the efficiency/practical effectiveness of the method is blue. If a method ends up with a red, green or blue color, it lacks benefit in the other two attributes. In addition to the final red, blue and green color, a method can be characterized as white (a perfect combination of all of them), magenta, cyan, yellow, colorless/gray and black. Magenta, cyan and yellow indicate some good combinations of many basic characteristics. The model is a flexible tool that can be adjusted, changed or enlarged. Currently, there are very few such transparent, holistic assessments available to the analytics community. The originators of this model point out that the concept does not have to apply only to analytical methods but can also be applied to various other scientific processes. In the absence of suitable tools for this type of evaluation, I ask the reviewer of the manuscript to be able to apply this suggestion to improve future publications that directly treat analytical method optimization procedures.
I hope that the comments presented will help improve the article.
Thank you for all the comments and suggestions of the Reviewer of the paper regarding the manuscript and your time.

Reviewer 2 Report
This study describes the presence of 15 polycyclic aromatic hydrocarbons (PAHs) in smoked and non-smoked cheeses purchased in Poland for safety control.
I believe that the contribution of this study is limited, as this type of analysis should be routine in a public control laboratory, and it refers only to Polish producers.
However, this type of study is necessary to know the magnitude of harmful compounds that may be present in this type of food, and to ensure that PAHs in food are kept at a level that does not endanger the health of consumers.
I believe the manuscript is suitable for publication in Molecules, but some issues need to be addressed first.
Title: I advise changing ¨PAH¨ to ¨Polycyclic Aromatic Hydrocarbons¨, as this term (PAH) is not commonly known. I would also delete ¨with fluorescence detection¨.... I believe it is of no relevance or value, unless it is justified as a faster and cheaper technique.
Introduction: The authors adequately explain the relevance of the problem and set out coherent and practical objectives.
Results and Discussion: This section is justified with references and current regulations.
Materials and Methods: This section is described in detail. Although the authors refer to Anastasio et al. [35], with modifications.....Have these modifications helped to improve the method?......Change from isocratic (Anastasio) to gradient.... If so, they could mention it in the manuscript. All improvements and contributions in any related aspect are important.
Conclusions: They are in line with the stated objectives, although I think they are somewhat extensive. I recommend summarising at the beginning, and at the end making some suggestions to reduce the problem posed.
Author Response
Comments and Suggestions for Authors
This study describes the presence of 15 polycyclic aromatic hydrocarbons (PAHs) in smoked and non-smoked cheeses purchased in Poland for safety control.
I believe that the contribution of this study is limited, as this type of analysis should be routine in a public control laboratory, and it refers only to Polish producers.
However, this type of study is necessary to know the magnitude of harmful compounds that may be present in this type of food, and to ensure that PAHs in food are kept at a level that does not endanger the health of consumers.
I believe the manuscript is suitable for publication in Molecules, but some issues need to be addressed first.
- Title:I advise changing ¨PAH¨ to ¨Polycyclic Aromatic Hydrocarbons¨, as this term (PAH) is not commonly known. I would also delete ¨with fluorescence detection¨.... I believe it is of no relevance or value, unless it is justified as a faster and cheaper technique.
The title has been improved according to the suggestion.
- Introduction:The authors adequately explain the relevance of the problem and set out coherent and practical objectives.
- Results and Discussion: This section is justified with references and current regulations.
- Materials and Methods:This section is described in detail. Although the authors refer to Anastasio et al. [35], with modifications.....Have these modifications helped to improve the method?......Change from isocratic (Anastasio) to gradient.... If so, they could mention it in the manuscript. All improvements and contributions in any related aspect are important.
Due to the fact that any chromatographic analysis requires the prior setting of appropriate chromatographic conditions (so-called method development) for a given analytical problem using the available apparatus, optimization of chromatographic separations was carried out based on the Anastasio et al. [35] methodology. This was based on concern for achieving adequate separation (resolution Rs>1.5), reasonable analysis time, acceptable pressure for instrument operation, minimum peak area relative to the maximum signal-to-noise ratio, robustness of the method, i.e., no changes during successive separations repeated with small changes in experimental conditions, and the minimum time and effort required to refine the method. The procedure aimed at determining the optimal composition of the mobile phase was carried out using gradient elution (i.e., under conditions where the composition of the mobile phase changes during the course of the analysis) - the so-called gradient approach. The final choice of eluent composition ensured an adequate retention range, satisfactory resolution and analysis time. Changes in the composition of the mobile phase were the simplest way of optimization.
- Conclusions:They are in line with the stated objectives, although I think they are somewhat extensive. I recommend summarising at the beginning, and at the end making some suggestions to reduce the problem posed.
The conclusions chapter has been improved according to the suggestion.
Thank you for all the comments and suggestions of the Reviewer of the paper regarding the manuscript and your time.

Round 2
Reviewer 1 Report
Dear Authors,
Thank you for your meticulous consideration of my comments. The paper has improved substantially and, in my opinion, is suitable for publication after minor revision.
1. RSD expressed as a percentage is the coefficient of variation (CV).
2. Figure 1. If possible, please add axis titles with units.
Author Response
COVER LETTER - For submission of manuscript
Date: 11.10.2022
Journal name: Molecules
Article type: Research Article
I am enclosing our manuscript entitled. "Evaluation of PAH in smoked cheeses made in Poland by HPLC method" for publication in the journal Molecules after making the necessary corrections according to the reviewers' comments and suggestions marked in the text of the paper, for which we thank you. Our Research Project was sponsored by the Minister of Education and Science in Poland under the programme entitled "Regional Excellence Initiative" for 2019-2023, with grant number Project No. 010/RID/2018/19, grant amount 12,000,000 PLN.
Magdalena Polak-Śliwińska
(Signature of corresponding author on behalf of all authors)
_______________________________________________________________________________
Comments and Suggestions for Authors - Reviewer 1
Dear Authors,
Thank you for your meticulous consideration of my comments. The paper has improved substantially and, in my opinion, is suitable for publication after minor revision.
- RSD expressed as a percentage is the coefficient of variation (CV).
We agree with the reviewer's comment, the correction has made in the text of the manuscript.
- Figure 1. If possible, please add axis titles with units.
Thank you very much for your suggestion. Axis titles with units have been presented more precisely.
Thank you for all the reviewer's comments and suggestions and for your time with regard to the manuscript under review, which resulted in a significant improvement of the manuscript.
